# Human low-density lipoprotein receptor plays an important role in hepatitis B virus infection

Yingying Li[1,2], Guangxiang Luo[1]*

**1** Department of Microbiology, University of Alabama at Birmingham School of Medicine, Birmingham, Alabama, United States of America, **2** Department of Microbiology, Peking University Health Science Center School of Basic Medical Sciences, Beijing, China

* gluo@uab.edu

**Data Availability Statement:** All relevant data are within the manuscript and its Supporting Information files.

**Funding:** This work was partially supported by NIH grants R01DK125734 and R01AI151007 to GL.

## Abstract

Hepatitis B virus (HBV) chronically infects more than 240 million people worldwide, resulting in chronic hepatitis, cirrhosis, and hepatocellular carcinoma. HBV vaccine is effective to prevent new HBV infection but does not offer therapeutic benefit to hepatitis B patients. Neither are current antiviral drugs curative of chronic hepatitis B. A more thorough understanding of HBV infection and replication holds a great promise for identification of novel antiviral drugs and design of optimal strategies towards the ultimate elimination of chronic hepatitis B. Recently, we have developed a robust HBV cell culture system and discovered that human apolipoprotein E (apoE) is enriched on the HBV envelope and promotes HBV infection and production. In the present study, we have determined the role of the low-density lipoprotein receptor (LDLR) in HBV infection. A LDLR-blocking monoclonal antibody potently inhibited HBV infection in HepG2 cells expressing the sodium taurocholate cotransporting polypeptide (NTCP) as well as in primary human hepatocytes. More importantly, small interfering RNAs (siRNAs)-mediated knockdown of LDLR expression and the CRISPR/Cas9-induced knockout of the LDLR gene markedly reduced HBV infection. A recombinant LDLR protein could block heparin-mediated apoE pulldown, suggesting that LDLR may act as an HBV cell attachment receptor via binding to the HBV-associated apoE. Collectively, these findings demonstrate that LDLR plays an important role in HBV infection probably by serving as a virus attachment receptor.

## Author summary

Requirement of multiple cell surface receptors and co-receptors for efficient virus infection is exemplified by human immunodeficient virus (HIV) and hepatitis C virus (HCV). In the case of HBV, expression of the NTCP receptor alone in human and murine hepatocytes converted HBV susceptibility albeit at low levels. Recent identification of the glypican 5 (GPC5) and epidermal growth factor receptor (EGFR) as HBV infection-promoting factors suggests that efficient HBV infection requires multiple cell surface molecules as virus attachment and post-attachment receptors. Here, we provide substantial evidence demonstrating that another cell surface receptor LDLR plays an important role in HBV infection. Downregulation of LDLR expression significantly lowered HBV infection,

The funders had no role in study design, data collection and analysis, decision to publish, or preparation of the manuscript.

**Competing interests:** The authors have declared that no competing interests exist.

whereas its upregulation promoted HBV infection. The levels of LDLR expression correlated with HBV cell attachment, suggesting that it serves as an HBV cell attachment receptor. The inhibition of heparin-mediated apoE pulldown by a purified LDLR suggested that LDLR promotes HBV infection probably through its binding to HBV-associated apoE. It is warranted to further determine whether other LDLR family members also play a role in HBV infection.

## Introduction

Hepatitis B virus (HBV) chronically infects more than 240 million people worldwide [1], resulting in common liver diseases such as hepatitis, liver fibrosis, cirrhosis, and hepatocellular carcinoma (HCC). Although prophylactic HBV vaccine has greatly reduced the number of new HBV infections and HCC cases, it does not offer therapeutic benefit to the hundreds of million people chronically infected with HBV. The current standard antiviral therapies consisting of interferon (IFN) and/or nucleoside analogs (NAs) can effectively suppress HBV replication but are not curative of hepatitis B unlike direct-acting antivirals (DAAs) for hepatitis C [2,3]. Chronic HBV infection is the leading cause of HCC, which is the most rapidly increasing cancer and ranks as the fifth most common cancer type and the third leading cause of cancer death worldwide [4,5]. Thus, HBV infection continues to pose a major threat to global public health.

HBV is the prototype member of the *Hepadnaviridae* family, consisting of small-enveloped DNA viruses with partially double-stranded DNA genomes of about 3.2 kb [6]. Upon HBV cell entry and uncoating, the viral polymerase protein attached to the viral DNA genome is removed in the cytoplasm, resulting in a deproteinized relaxed circular DNA (DP rcDNA). The DP rcDNA is subsequently transported into the nucleus and is converted to covalently closed circular DNA (cccDNA) [7,8]. The rcDNA to cccDNA conversion is likely carried out by cellular enzymes, including DNA polymerase, ligase, and enzymes involved in host DNA damage repair systems. However, the underlying molecular mechanism of cccDNA synthesis and maintenance in the nucleus remains largely unknown [9,10]. The cccDNA is the template for transcription of all viral RNAs, including mRNAs and a terminally redundant pregenomic RNA (pgRNA). The viral mRNAs and pgRNA encode seven proteins such as three different forms (L, M, and S) of envelope proteins (HBs), preCore (HBe precursor), core (HBc), polymerase (P), and X protein (HBx). The pgRNA and viral polymerase protein are encapsidated by the core protein to form nucleocapsids where reverse transcription of the pgRNA takes place. In the past, a great deal of new knowledge has been obtained regarding the molecular aspects of HBV DNA replication through studies with recombinant DNA approaches [6]. However, very little is known about the underlying molecular mechanisms of HBV infection, morphogenesis, and egress due largely to the lack of *bona fide* cell culture models of HBV propagation.

The discovery of sodium taurocholate cotransporting polypeptide (NTCP) as the HBV receptor has made it possible to develop more robust cell culture models of HBV infection [11], as reported by numerous groups including us [12–21]. Employing cell culture systems of HBV production and infection [17], we have recently found that human apolipoprotein E (apoE) is enriched on the HBV envelope and is important for efficient HBV infection and production [22]. An apoE-specific monoclonal antibody (mAb23) was able to not only capture HBV but also potently block HBV infection. Silencing of apoE expression by small interfering RNA (siRNA) or knockout of apoE gene by CRISPR/Cas9 could efficiently reduce HBV

infection and production. Interestingly, apoE was previously found to also play a critical role in hepatitis C virus (HCV) cell attachment and morphogenesis via binding to hepatocyte surface receptors heparan sulfate proteoglycans (HSPGs) and viral proteins NS5A and E1/E2, respectively, as demonstrated by several independent studies [23–25]. ApoE is known to play a central role in the transport, metabolism, and homeostasis of cholesterol and apoE-containing lipoproteins by serving as a critical ligand for the low-density lipoprotein receptor (LDLR) [26–28]. Therefore, we believe that apoE promotes HBV infection through binding to cell surface receptors such as LDLR and HSPGs.

HSPGs are implicated in HBV cell attachment, as suggested by previous studies [29,30]. Removal of HSPGs from cell surface by treatment of cells with heparinase conferred less susceptibility to HBV infection [29]. Similarly, heparin was shown to block HBV infection in cell culture [30]. One of the HSPG core proteins, glypican 5 (GPC5), was found to be an HBV entry factor [31,32]. These previous studies suggest that HSPGs likely serve as HBV attachment receptors. However, the role of LDLR in HBV infection has not been experimentally examined. In the present study, we sought to determine the role of LDLR in HBV infection. Strikingly, a LDLR-specific monoclonal antibody (C7) potently blocked HBV infection in the NTCP-expressing HepG2 cell line and primary human hepatocytes (PHHs). Likewise, silencing of LDLR expression and the CRISPR/Cas9-induced knockout of the LDLR gene efficiently lowered HBV infection but did not affect HBV DNA replication. Interestingly, a recombinant LDLR protein was able to block apoE pull-down. Collectively, these findings suggest that LDLR promotes HBV infection probably through its interaction with apoE enriched on the HBV envelope.

## Results

### Efficient blockade of HBV infection by a LDLR-specific monoclonal antibody

We have previously found that human apoE is enriched in HBV envelope and plays an important role in HBV infection although its mechanism of action has not been defined [22]. It is known that both HSPGs and LDLR family proteins serve as the apoE-binding receptors [28]. HSPGs was previously found to facilitate both HBV and HCV cell attachment [24,29,30,33–35]. To determine whether LDLR plays a role in HBV infection, we initially tested a LDLR-blocking monoclonal antibody (C7) using our previously described HepG2$^{NTCP}$ cell culture model of HBV infection [17]. HepG2$^{NTCP}$ cells were efficiently infected with HBV, as demonstrated by increasing levels of HBcAg expression up to 5 days after infection, which were determined by both Western blot analysis (S1A Fig) and IFA (S1B Fig). Thus, this robust HBV cell culture model was used in this study. HBV was mixed with varying amounts (0, 0.4, 2, and 10 μg/mL) of C7 prior to its infection of HepG2$^{NTCP}$ cells. The effect of C7 on HBV infection was subsequently determined by quantifying the levels of HBV DNAs and proteins in the cell and cell culture supernatant. The LDLR-specific monoclonal antibody C7 was found to potently block HBV infection, as determined by dose-dependent reduction of HBcAg and HBV cccDNA in cells as well as HBeAg and HBV DNA in supernatants (Fig 1). The levels of HBcAg, HBV cccDNA, and HBeAg were lowered by about 60% at a concentration of 10 μg/mL of C7 compared to no antibody control or 10 μg/mL of a normal mouse IgG (nmIgG) (Fig 1A, 1B, and 1C). The level of HBV DNA in the supernatant was 5-folds lower in the presence of 10 μg/mL of C7 compared to the control without C7 or 10 μg/mL of nmIgG (Fig 1D). To determine the physiological importance of LDLR in HBV infection, we used primary human hepatocytes (PHHs) for HBV infection. HBV was incubated with increasing amounts (0, 0.4, 2, and 10 μg/mL) of C7 during infection. Again, C7 efficiently blocked HBV infection in PHHs, resulting in 57%, 70%, and 88% reduction of HBcAg in the cell at concentrations of 0.4, 2, and 10 μg/mL of C7 (Fig 2A). The level of HBeAg in the supernatants were also

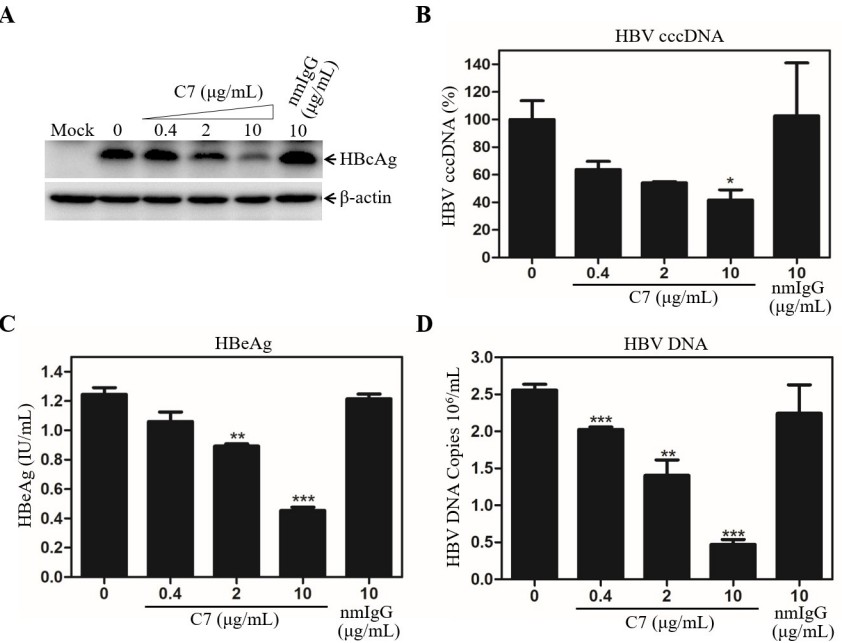

**Fig 1. Blockade of HBV infection by a LDLR-specific monoclonal antibody.** HepG2$^{NTCP}$ cells seeded in 24-well plates were infected with HBV in the presence of 4% PEG and increasing concentrations (0, 0.4, 2, and 10 μg/mL) of the LDLR monoclonal antibody (C7). A normal mouse IgG (nmIgG) was used as a negative control. After 12-h infection at 37˚C, the HBV-infected cells were washed with PBS and incubated with DME/F12 containing 3% FBS, 1% DMSO, 5 μg/mL hydrocortisone (HC), and corresponding concentrations of C7 or nmIgG for 4 days. Cell lysates were collected for detection of HBV core antigen (HBcAg) by Western blot using β-actin as a house-keeping gene control (**A**). HBV cccDNA in the cells was extracted with the *Hirt* method and treated with exonucleases I, III, and T5, as previously described [22]. The levels of HBV cccDNA were determined by a Real-Time PCR method (**B**). The levels of HBeAg in the supernatants were quantified by the previously described chemiluminescence immunoassay (**C**). The levels of HBV genomic DNA in the supernatants were also quantified by a Real-Time PCR method (**D**). The average levels of HBV cccDNA, HBeAg, and HBV DNA were obtained from three independent experiments and plotted against C7 concentrations.*P < 0.05, **P < 0.01, ***P < 0.001.

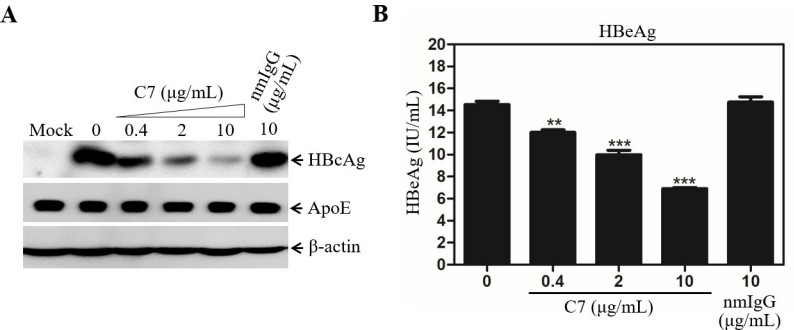

**Fig 2. C7 neutralization of HBV infectivity in PHH.** PHHs seeded in a 24-well plate were infected with HBV in the presence of 4% PEG and increasing concentrations (0, 0.4, 2, and 10 μg/mL) of C7 using nmIgG (10 μg/mL) as a negative control. After 12-hrs incubation at 37˚C, the HBV-infected PHHs were washed with PBS and incubated in Power primary HEP medium with the corresponding concentrations of C7 or nmIgG used during HBV infection. At 4-days post-infection (p.i.), cells were lysed with RIPA buffer. The levels of HBcAg were determined by Western blot analysis (**A**). The levels of HBeAg in the supernatants were quantified by a chemiluminescence immunoassay (**B**). Average levels of HBeAg in the supernatants obtained from triplicate repeats were calculated and plotted against the concentrations of C7. **P < 0.01, ***P < 0.001.

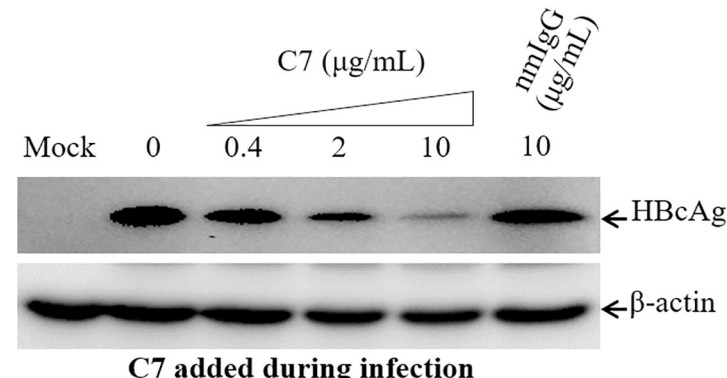

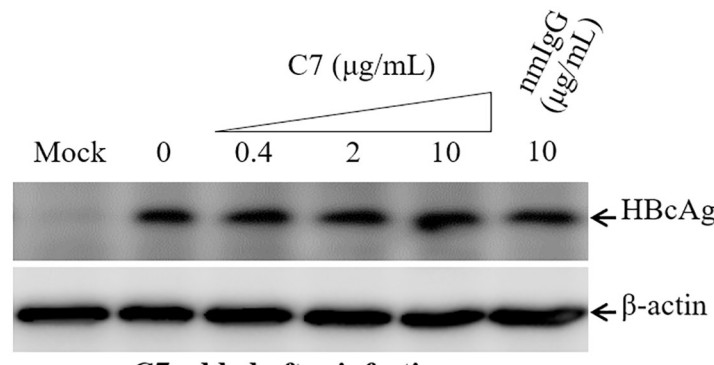

**Fig 3. Comparison of HBV neutralization activity of C7 added during or after HBV infection.** HepG2^NTCP cells in a 24-well plate were incubated with HBV for 12-hrs in the presence of 4% PEG and increasing concentrations (0, 0.4, 2, and 10 μg/mL) of C7 or nmIgG (10 μg/mL) added during (**A**) or after (**B**) HBV infection. HBV-infected HepG2^NTCP cells were incubated with DME/F12 medium containing 3% FBS, 1% DMSO, and 5 μg/mL hydrocortisone (HC) at 37˚C for 4 days. Cell lysates were collected for detection of HBcAg by Western blot. β-actin was used as a control for protein normalization. Antibody concentrations are indicated on the top. Cell lysate without HBV infection was used as a mock control.

proportionally decreased by up to greater than 50% (Fig 2B). The blockade of HBV infection by C7 was also determined when added during or after HBV infection (Fig 3). C7 blocked HBV infection only when added during infection, resulting in proportional reduction of HBcAg by up to 65% (Fig 3A). However, the levels of HBcAg remained unchanged when C7 was added after HBV infection (Fig 3B). HepG2^NTCP cells do not support HBV propagation, which explains why C7 did not affect the levels of HBcAg when added after HBV infection. Taken together, these results suggest that LDLR plays an important role in HBV infection.

## Impairment of HBV infection by down-regulation of LDLR expression

To further validate the importance of LDLR in HBV infection, we have determined the effects of down-regulation of LDLR expression on HBV infection. Initially, a smartpool of LDLR-specific siRNAs (siLDLR) was used to silence LDLR expression in HepG2^NTCP cells, followed by HBV infection. A non-specific control siRNA (siNSC) was included in the experiments. Unlike siNSC, siLDLR was able to reduce LDLR expression in a dose-dependent manner to an

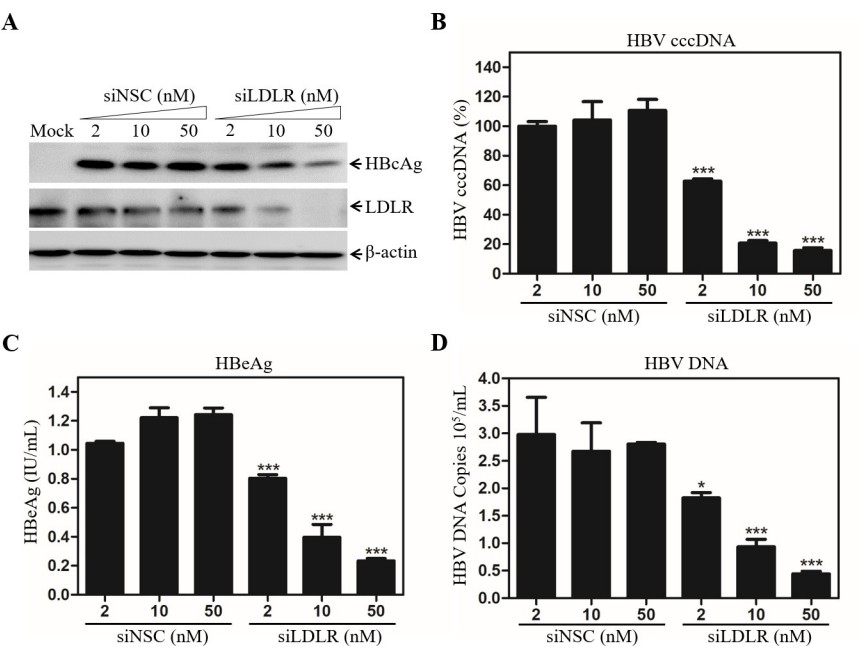

**Fig 4. Reduction of HBV infection by LDLR-specific siRNAs.** HepG2^NTCP cells were transfected with a Smartpool of LDLR-specific siRNAs at varying concentrations (0, 2, 10, 50 nM) using RNAiMax (Invitrogen) as described in our previous work [22]. A non-specific control siRNA (siNSC) was used as a control. At 2-days post-transfection (p.t.), cells were infected with HBV in the presence of 4% PEG at 37°C for 12 hrs. At 4-days p.i., the levels of LDLR and HBcAg in the cell were determined by Western blot using specific antibodies (**A**). HBV cccDNA in cells was extracted and quantified by a Real-Time PCR method (**B**), as described previously [22]. The levels of HBeAg in the supernatants were measured by a chemiluminescence immunoassay (**C**). The levels of HBV DNA in the supernatants were quantified by a Real-Time PCR method (**D**). Average values of triplicates were calculated and plotted against siRNAs concentrations. *P < 0.05, ***P < 0.001.

undetectable level at 50 nM as shown by Western blot analysis (Fig 4A). As a result, the levels of HBcAg in the HBV-infected HepG2^NTCP cells were proportionally decreased by 15%, 48% and 74% at 2, 10, and 50 nM of siRNAs (Fig 4A). Similarly, the level of HBV cccDNA were lowered by 40%, 80%, and 85% at 2 nM, 10 nM and 50 nM of LDLR-specific siRNAs (Fig 4B). HBeAg and HBV DNA in the cell culture supernatants were reduced by greater than 80% at 50 nM siLDLR concentration compared to those in the siNSC-transfected cells (Fig 4C and 4D). The reduction of HBV infection by silencing LDLR expression was also confirmed by HBcAg immunostaining of HBV-infected cells (S2 Fig). These results demonstrate that LDLR is required for efficient HBV infection.

To further verify the importance of LDLR in HBV infection, we used the CRISPR/Cas9 technology to make stable LDLR gene-knockout (LDLR^-/-) HepG2^NTCP cell lines. Upon selection with blasticidin, we have obtained a number of cell clones. Three independent clones were chosen for the subsequent experiments, which contain 20, 7, and 10 nucleotide deletions around the sgRNA targeting region, respectively, as determined by DNA sequence analysis (Fig 5A). As expected, LDLR was undetectable among these stable LDLR^-/- HepG2^NTCP cell lines, as shown by Western blot analysis (Fig 5A). The susceptibility of these LDLR^-/- cell lines was determined by an HBV infection assay. Strikingly, LDLR gene knockout resulted in 80% to 90% reduction of HBcAg and HBV cccDNA in the HBV-infected cells (Fig 5A and 5B). HBeAg and HBV DNA in LDLR^-/- HepG2^NTCP cell culture supernatants were greater than 85% lower than those in the parental (w.t.) cells (Fig 5C and 5D). These results confirm that LDLR plays an important role in HBV infection.

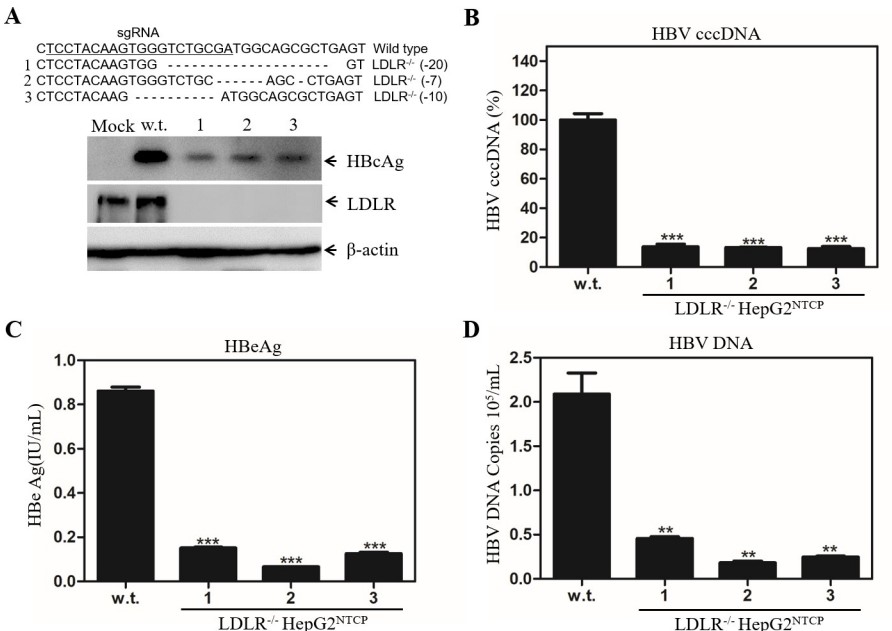

**Fig 5. Impairment of HBV infection in the LDLR-knockout HepG2<sup>NTCP</sup> cells.** HepG2[NTCP] cells were transduced with a lentivirus (LentiCRISPRv2-Blast/LDLR-sgRNA) expressing a Cas9 and LDLR-specific sgRNA. Upon selection with blasticidin (5μg/mL), stable cell clones were picked up and expanded. Genomic DNAs were extracted from stable cell lines and were used for amplification of the LDLR gene spanning the sgRNA-target region by PCR. PCR products were subjected to DNA sequence analysis. The nucleotide deletions of the LDLR gene from three stable cell lines are shown on the top of A. The deficiency of LDLR expression in the LDLR[-/-] cell lines was validated by Western blot analysis. The LDLR-deficient cell lines were infected with HBV in a parallel comparison with the parental HepG2[NTCP] cells (w.t.). At 4-days p.i., the levels of LDLR and HBcAg were determined by Western blot using β-actin as a loading control (**A**). The levels of HBV cccDNA in the parental and LDLR-deficient cells were quantified by a Real-Time PCR method. The relative levels of cccDNA were converted to percentage (%) of that in parental cells which was considered 100% (**B**). The levels of HBeAg in the supernatants were quantified by a chemiluminescence immunoassay (**C**). The levels of HBV genomic DNA in the supernatants were quantified by Real-Time PCR and shown as genomic DNA copy numbers (**D**). Data were presented as average values ± standard deviation obtained from three independent experiments. **P < 0.01, ***P < 0.001.

The question arose whether the defective HBV infection in the LDLR[-/-] HepG2[NTCP] cells was due to off-target effect associated with CRISPR/Cas9 gene-editing system. To exclude this possibility, we sought to determine if the defective HBV infection in the LDLR-null cells could be restored by ectopic LDLR expression. The stable LDLR[-/-] cell line 3 was used for restoration experiments. The human LDLR-expressing plasmid pCMV6-XL4-hLDLR was transfected into LDLR[-/-] HepG2[NTCP] cells, which were subsequently infected with HBV at 48h post-transfection. The levels of HBcAg and HBV cccDNA in the cell were determined by WB and qPCR, respectively. Results from these experiments showed that the defective HBV infection could be fully restored by ectopic LDLR expression as determined by the levels of HBcAg (Fig 6A) and HBV cccDNA (Fig 6B).

## Enhancement of HBV infection by LDLR overexpression

Down-regulation of LDLR expression caused a significant reduction of HBV infection. Over-expression of LDLR in the parental HepG2[NTCP] cells appeared to further enhance HBV infection (Fig 6A and 6B). To confirm if HBV infection could be enhanced by overexpression of LDLR in HepG2[NTCP] cells, increasing amounts of the LDLR-expressing vector pCMV6-XL4-hLDLR were transfected into HepG2[NTCP] cells, followed by HBV infection at 48h p.t. Interestingly, the levels of HBcAg were significantly increased in proportion to the levels of

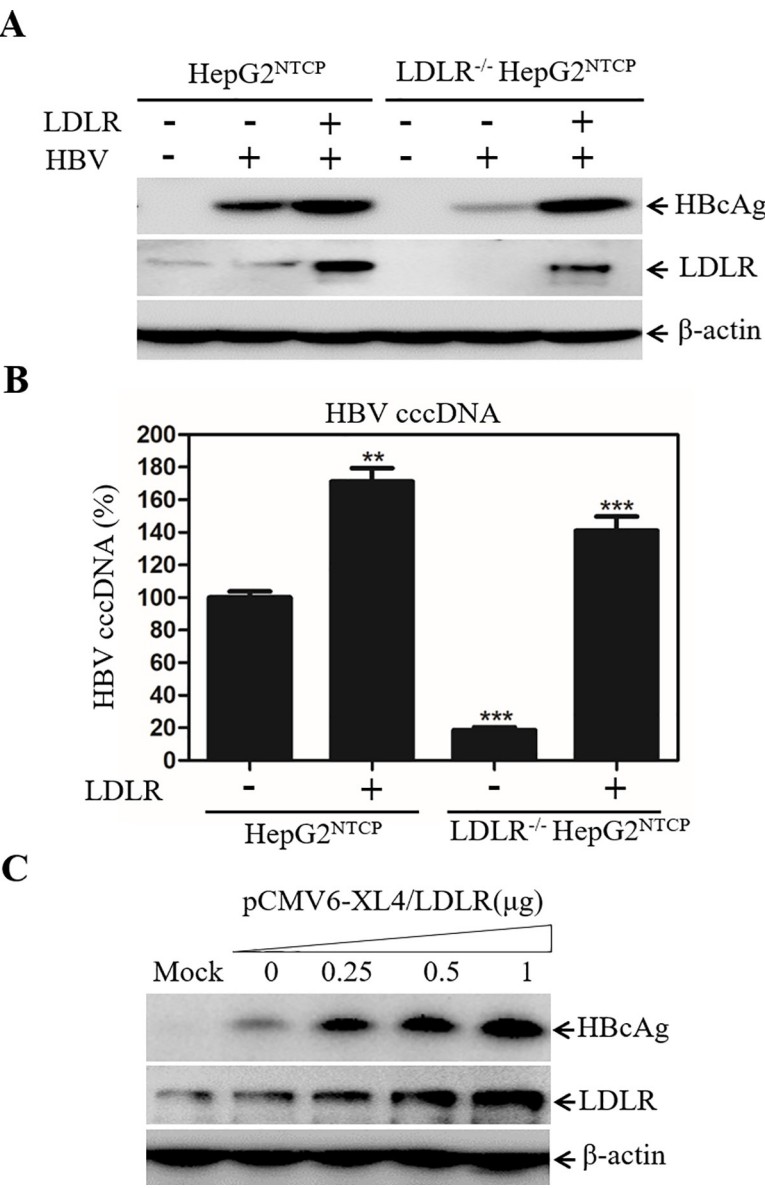

**Fig 6. Restoration and enhancement of HBV infection by ectopic LDLR overexpression.** Parental and LDLR-knockout (LDLR[-/-]) HepG2[NTCP] cells in 24-well cell culture plates were transfected with a LDLR-expressing vector (pCMV6-XL4-hLDLR). At 48h p.t., cells were infected with HBV. At 4d p.i., cell lysates were collected for detection of HBcAg and LDLR by Western blot (**A**). The levels of HBV cccDNA in the cell were quantified by a Real-Time PCR method. The relative levels of HBV cccDNA were obtained from three repeats and converted to the percentage (%) of control considering 100% of the HBV cccDNA in the parental HepG2[NTCP] cells (**B**). **P < 0.01, ***P < 0.001. Parental HepG2[NTCP] cells in a 24-well plate were also transfected with a total of 1 μg plasmid DNA consisting of varying amounts (0, 0.25, 0.5, and 1 μg) of the LDLR-expressing plasmid DNA and the vector DNA used for keeping DNA at constant 1 μg. At 48-h p.t., cells were infected with HBV at 37°C for 12 h, followed by culturing in DME/F12 medium containing 3% FBS, 1% DMSO, and 5 μg/mL hydrocortisone for 4 days. The levels of HBcAg in the cell were determined by Western blot (**C**).

LDLR expression (Fig 6C). Clearly, HBV infection was well correlated with the levels of LDLR expression. These results demonstrate that the level of LDLR expression determines HBV infection efficiency, consistent with the findings obtained from the above-described down-regulation of LDLR expression (Figs 4 and 5).

## Down-regulation of LDLR expression did not affect HBV DNA replication

To exclude the possible effect of LDLR deficiency on HBV DNA replication, an infectious HBV plasmid DNA (pCMV-HBV) was transfected into LDLR$^{-/-}$ HepG2$^{NTCP}$ cell lines. HBV DNA replication was determined by quantifying the levels of HBcAg, HBV cccDNA, and HBV DNA in the HBV DNA-transfected cells after treatment of DNA samples with the restriction enzyme DpnI (to remove plasmid DNA) and exonucleases I, III, and T5 to degrade HBV rcDNA [36]. The levels of HBcAg (Fig 7A), HBV cccDNA (Fig 7B), and replicated HBV DNA (Fig 7C) remained unchanged between the parental and the LDLR-null HepG2$^{NTCP}$cells. These data suggest that the deficiency of LDLR expression did not affect HBV DNA replication.

## LDLR serves as an HBV cell attachment receptor via binding to the HBV-associated apoE

The LDLR-blocking monoclonal antibody C7 inhibited HBV infection when added during but not after HBV infection, suggesting its action occurred at a very early step of HBV infection. To define the underlying molecular mechanism of LDLR in the mediation of HBV infection, we carried out an HBV cell attachment assay in the absence and presence of the LDLR monoclonal antibody C7. HBV was incubated with HepG2$^{NTCP}$ cells for 6 hours with or without C7. The cell-bound HBV DNA was extracted for qPCR quantification upon removal of unbound HBV after extensive washing with PBS. Consistent with its potency in HBV infection, C7 also blocked HBV cell attachment, resulting in about 80% reduction of HBV DNA at 10 μg/mL (Fig 8A). The levels of HBV DNA were also lowered by 70–80% in the LDLR$^{-/-}$ HepG2$^{NTCP}$ cell lines (Fig 8B). These results suggest that LDLR may function as an HBV attachment receptor. However, HBV entry to cells could not be excluded when HepG2$^{NTCP}$ cells were used for HBV cell attachment assay. To further confirm the importance of LDLR in HBV attachment, the parental HepG2 cells (without NTCP overexpression) were used for down-regulation of LDLR expression by specific siRNAs, followed by HBV attachment. HepG2 cells were transfected with LDLR-specific siRNAs along with a non-specific siRNA control. As expected, LDLR-specific siRNAs silenced LDLR expression to undetectable levels (Fig 9A). As a result, HBV cell attachment was significantly reduced by about 75% at 25 nM of siRNA (Fig 9B). These results demonstrate that LDLR does play an important role in HBV cell attachment.

LDLR is a known apoE-binding receptor like HSPGs [27]. We have previously demonstrated that human apoE is enriched on the HBV envelope and promotes HBV infection [22]. To determine specific interactions between LDLR and apoE, we used a heparin-mediated apoE pull-down assay to examine the blockade of heparin and apoE interaction by LDLR. The heparin-conjugated agarose beads were incubated with apoE in the absence or presence of a purified human LDLR. Interestingly, LDLR efficiently blocked apoE-binding to heparin (Fig 10A), resulting in a dose-dependent reduction of apoE pull-down as shown by the ratios between heparin-bound and unbound apoE (Fig 10B). These results suggest that LDLR may mediate HBV cell attachment through binding to apoE enriched in the HBV envelope.

## Discussion

A number of previous studies have demonstrated that NTCP is the cell surface receptor essential for HBV infection. However, HBV infection in the NTCP-expressing human and murine hepatocytes is very inefficient and relies on the presence of PEG and high HBV inoculum unlike HBV transmission in humans. Neither were NTCP-transgenic mice susceptible to HBV

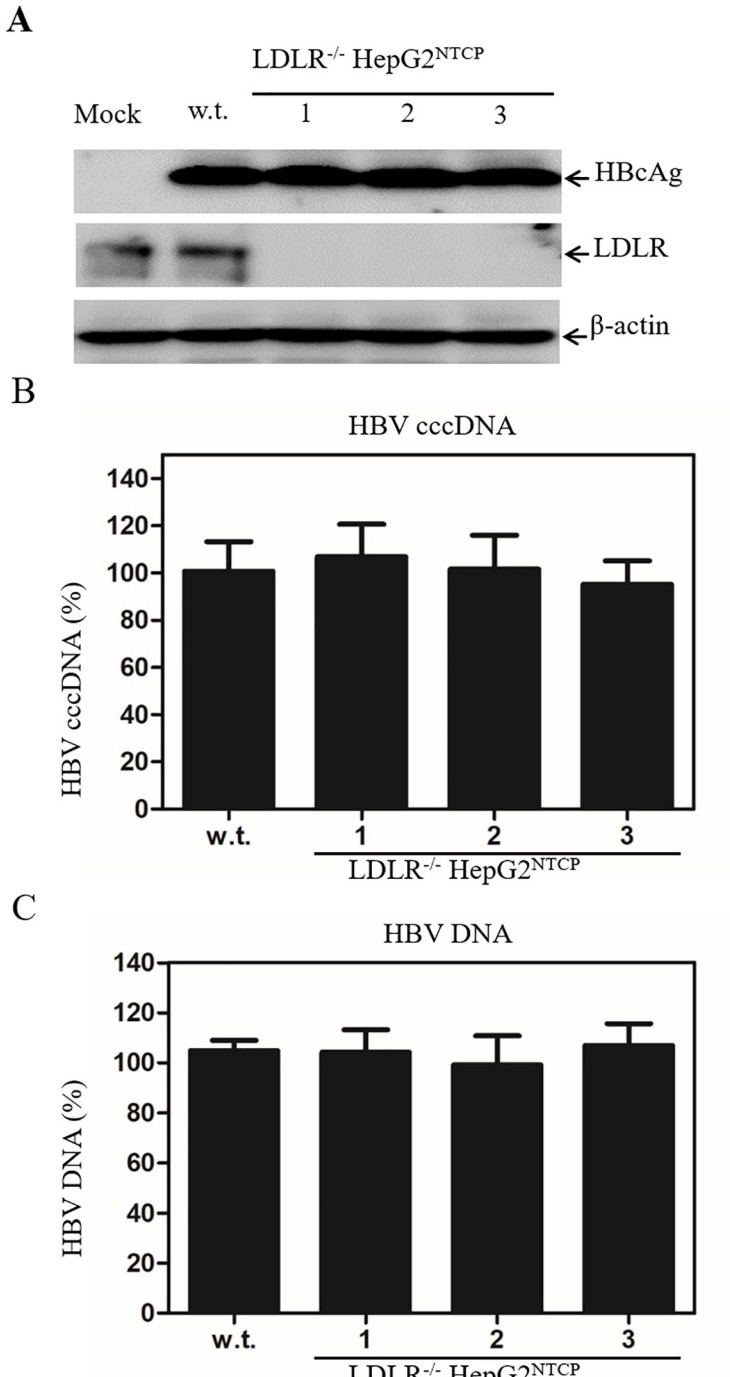

**Fig 7. Effects of LDLR gene knockout on HBV DNA replication.** The LDLR$^{-/-}$ HepG2$^{NTCP}$ cell lines in 24-well plates were transfected with 1 μg of infectious HBV DNA plasmid (pCMV-HBV) that was previously described [59]. At 48 h p.t., cell lysates were collected and DNA was extracted with a QIAGEN DNA isolation kit. The levels of HBcAg (**A**) were determined by Western blot. HBV cccDNA was isolated using the Hirt method and total HBV DNA was extracted with a QIAGEN DNA isolation kit. Possible HBV plasmid DNA was removed by treating DNAs with the restriction enzyme DpnI. HBV rcDNA was digested by treatment of DNA sample with exonuclease I, III, and T5. The levels of HBV cccDNA (**B**) and DNA (**C**) were determined by qPCR methods. The levels of HBV cccDNA and DNA are shown as percentage (%) of control considering 100% of HBV cccDNA and DNA in the parental HepG2$^{NTCP}$ cells (w.t.).

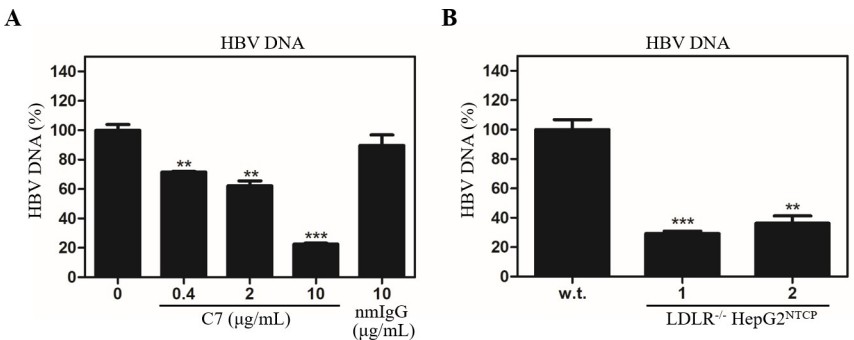

**Fig 8. Inhibition of HBV cell attachment by C7 and LDLR gene knockout.** HepG2$^{NTCP}$ cells in 24-well cell culture plates were incubated with HBV in the presence of increasing concentrations (0, 0.4, 2, and 10μg/mL) of LDLR-specific monoclonal antibody C7 or 10 μg/mL nmIgG at 37˚C for 6 h. Unbound HBV was removed by washing cells with 1xPBS three times. Cell-bound HBV DNA was extracted with a QIAGEN DNA isolation kit, followed by real-time PCR quantification (**A**). HBV cell attachment to parental and LDLR$^{-/-}$ HepG2$^{NTCP}$ cells were compared after incubation with HBV at 37˚C for 6 h. Upon removal of unbound HBV, HBV DNA in the cell was quantified by a Real-Time PCR method (**B**). Average values calculated from three repeats are shown. $^{**}P < 0.01$, $^{***}P < 0.001$.

infection [37,38]. The lack of robust HBV infection in the NTCP-expressing hepatocytes and mice suggests additional cellular factors required for a *bona fide* HBV propagation. GPC5 was previously shown to promote HBV infection [32]. A more recent study revealed that epidermal growth factor receptor (EGFR) also serves as an HBV entry cofactor through a specific interaction with NTCP [39]. The identification of GPC5 and EGFR as HBV entry factors implies the existence of multiple cell surface receptors/coreceptors involved in HBV infection like HCV [40]. In the present study, we have found that another hepatocyte surface molecule LDLR plays an important role in HBV infection, as demonstrated by several lines of substantial evidence. First, an LDLR-specific monoclonal antibody (C7), which was previously shown to block lipoprotein uptake [41], could potently inhibit HBV infection in the NTCP-expressing HepG2$^{NTCP}$ cells (Fig 1). C7 blocked HBV infection only when it was added during but not after HBV infection (Fig 3). More importantly, silencing of LDLR expression or knockout of the LDLR gene resulted in a remarkable reduction of HBV infection (Figs 4 and 5). The defective HBV infection in the LDLR-null HepG2$^{NTCP}$ cells could be fully restored by ectopic expression of LDLR (Fig 6). More significantly, the physiological importance of LDLR in HBV infection was demonstrated in PHHs using LDLR-specific monoclonal antibody C7, which potently blocked HBV infection (Fig 2). In contrast, down-regulation of LDLR expression did not affect HBV DNA replication, as determined by similar levels of HBcAg and HBV cccDNA upon transfection of an infectious HBV DNA into the LDLR-deficient HepG2$^{NTCP}$ cells (Fig 7). On the other hand, overexpression of LDLR in HepG2$^{NTCP}$ cells could significantly enhance HBV infection in a dose-dependent manner (Fig 6C). The positive correlation between HBV infection and the levels of LDLR expression demonstrates the importance of LDLR in HBV infection.

LDLR is a single chain transmembrane glycoprotein of 839 amino acids (aa), consisting of a large N-terminal extracellular domain (767aa), transmembrane region (22aa), and the C-terminal cytoplasmic tail (50aa). The N-terminal portion of LDLR contains seven cysteine-rich repeats of 40 residues that constitute the ligand-binding domain. Immediately after the cysteine-rich repeats is a 400-residue region consisting of two epidermal growth factor-like (EGF-like) repeats, a YWTD domain, and a third EGF-like repeat [42]. LDLR plays a critical role in lipid and cholesterol metabolism via its binding to apoB100 on the LDL and apoE on the chylomicron, very-low-density lipoprotein (VLDL), and high-density lipoprotein (HDL) particles.

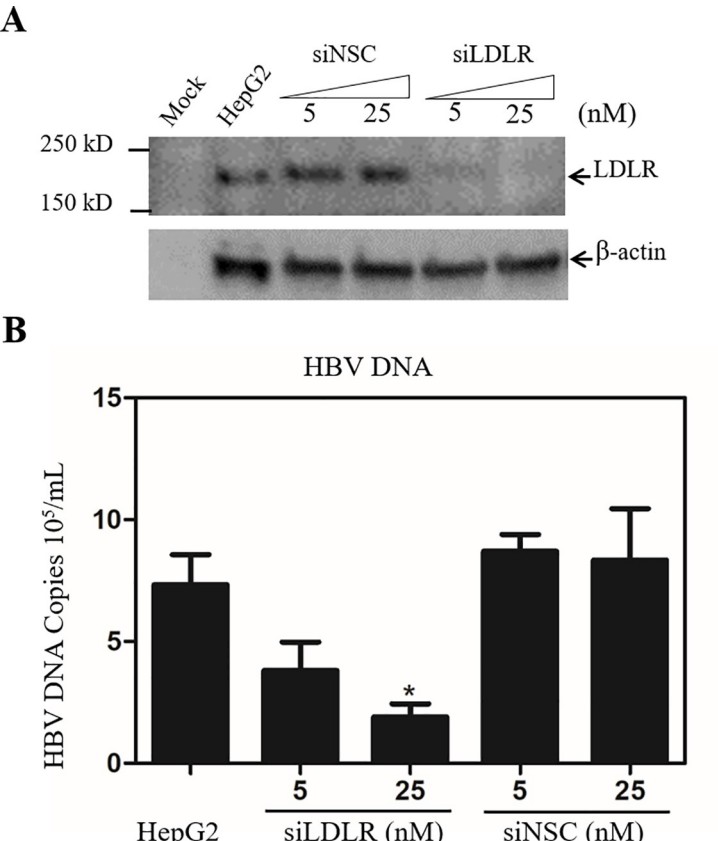

**Fig 9. Reduction of HBV cell attachment by siRNA-induced silencing of LDLR expression in HepG2 cells.** The LDLR-specific siRNAs (siLDLR) and a non-specific control siRNA (siNSC) were transfected into parental HepG2 cells in 24-well plates in the same way as in Fig 4. After 48 hrs post-transfection, cells were incubated with HBV at 37˚C for 12 h. The unbound HBV was removed by aspiration and washing with PBS three times. The cell-bound HBV DNA was extracted and quantified as described in Fig 8. Cells were also lysed in a RIPA buffer for determining the levels of LDLR by a Western blot analysis. **A**. Determination of LDLR expression. The levels of LDLR in siRNA-transfected HepG2 and mock control HepG2 were determined by Western blot using an LDLR-specific monoclonal antibody (HL-1). β-actin was used as an internal control. **B**. The levels of HBV DNA determined by a qPCR method. The genomic HBV DNA copies were calculated as average of three repeats. The concentrations of siRNA are indicated on the top (A) or bottom (B). Protein size markers are shown on the left.

In addition to its importance in the uptake of lipoprotein particles, LDLR has been found to serve as an infection-promoting factor for several viruses, including rhinovirus [43], vesicular stomatitis virus (VSV) [44], Rous sarcoma virus [45], and HCV [46]. In the case of rhinovirus, the ligand-binding cysteine-rich repeats of LDLR were found to bind to VP1 [47,48]. The underlying molecular mechanism of the LDLR-mediated VSV cell entry has not been defined. The role of LDLR in HCV infection was probably through an interaction with the host protein apoE that is enriched on the HCV envelope [49]. Circumstantial evidence derived from our current study suggests that LDLR may serve as an attachment receptor for HBV infection. The LDLR-specific monoclonal antibody C7 could efficiently block HBV attachment to HepG2$^{NTCP}$ cells, resulting in 80% reduction of the cell-bound HBV particles (Fig 8A). Like-wise, the knockout of LDLR gene decreased HBV cell attachment by about 70% (Fig 8B). Like HSPGs, LDLR is also a well-known receptor for apoE-binding, which mediates the influx of lipoproteins [50]. We have previously demonstrated that apoE is enriched in HBV envelope and is critical for efficient HBV infection [22]. We speculate that LDLR promotes HBV

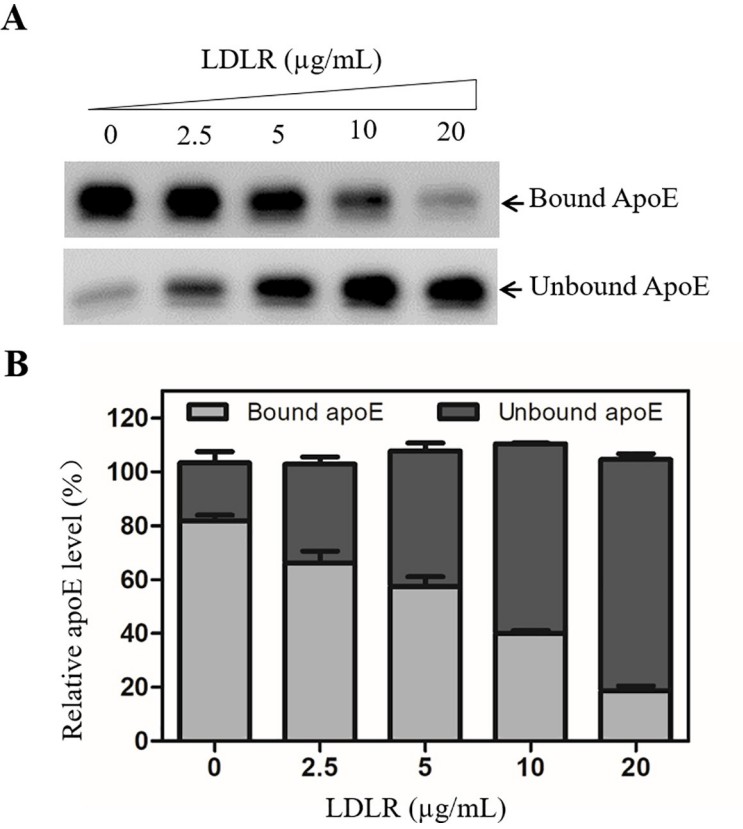

**Fig 10. Inhibition of the heparin-binding of apoE by a purified recombinant LDLR.** The cell culture supernatant of Huh7 cells, which contained high level of apoE, was mixed with heparin-conjugated agarose beads in the absence or presence of varying amounts (0, 2.5, 5, 10, 20 μg/ml) of a purified human LDLR. Upon washing with PBS, the heparin-bound (pulldown) and unbound apoE protein was determined by Western blot analysis using apoE-specific WuE4 monoclonal antibody (**A**). The percentage of heparin-bound and unbound apoE is plotted using average values obtained from three repeats (**B**).

infection likely through its interaction with the HBV-associated apoE. This possibility is supported by the finding that purified recombinant LDLR was able to block the heparin-mediated apoE pull-down *in vitro* (Fig 10). ApoE was also previously shown to be enriched on the HCV envelope, mediating HCV cell attachment [23,24,34,51,52]. However, we have not excluded a possible interaction between LDLR and the viral envelope protein HBsAg, which is warranted by future investigations.

Down-regulation of LDLR expression significantly impaired but did not diminish HBV infection, suggesting the involvement of other attachment receptors in HBV infection. Several previous studies suggested that HSPGs serve as HBV attachment receptors. Additionally, other members of the LDLR family may also play a role in HBV infection, including the VLDLR and LDLR-related proteins (LRPs). It was previously shown that the LDLR family members such as LDLR, VLDLR, LRP1, LRP5, and LRP8 can all bind to apoE [50,53–57]. It will be interesting to determine the importance of other LDLR family members besides LDLR in HBV infection. More significantly, the illustration of the underlying molecular mechanism of the LDLR family members in HBV infection will not only provide new knowledge about the cell surface attachment receptors in HBV infection but also novel targets and/or pathways for discovery and development of antiviral drugs towards the elimination of chronic hepatitis B.

## Materials and methods

### Cell culture

The NTCP-expressing HepG2 cell line (HepG2$^{NTCP}$) and the HBV-producing HepG2 cell line (HepAD38) were described previously [17,58]. They were cultured in DME/F12 medium supplemented with 10% fetal bovine serum (FBS), 100 U/mL penicillin, 100 U/mL streptomycin (Sigma), 1x MEM non-essential amino acids, and 1x sodium pyruvate, at 37˚C in a 5% $CO_2$ incubator. PHHs (lot#4405C) were purchased from Lonza and cultured according to the manufacturer's instruction except for the use of Power Primary HEP medium (Takara). HEK293T cells were maintained in DMEM medium containing 10% FBS. All cell culture flasks and plates were coated with 50 μg/mL rat tail collagen type I (Corning) prior to seeding of cells.

### Antibodies and reagents

The LDLR-blocking monoclonal antibody C7, which was described previously [41], was produced from a hybridoma obtained from ATCC (CRL-1691) and were purified through affinity chromatography by GenScript (Piscataway, NJ). Mouse anti-LDLR (clone HL1) monoclonal antibody was provided by Jin Ye [49]. A normal mouse IgG was purchased from Santa Cruz Biotechnology. HBcAg-specific mouse monoclonal antibodies (T2221 and C1-5) were purchased from Tokyo Future Style and Santa Cruz Biotechnology, respectively. Goat anti-mouse was from Human β-actin monoclonal antibody (AC15) and protein A and G agarose beads were from Sigma-Aldrich. Mouse anti-NTCP monoclonal antibody was described previously [17]. ApoE-specific monoclonal antibody WuE4 (ATCC) was produced in the lab as described previously [51]. HRP-conjugated goat anti-mouse antibody was from Cell Signaling. Protein assay dye reagent and Clarity Max Western blotting ECL substrate were purchased from Bio-Rad. The LDLR-specific siRNAs and a nonspecific control (NSC) siRNA were synthesized by Dharmacon. Exonucleases I (Exo I), III (Exo III), and T5, as well as Taq 5× master mix were from New England Biolabs. HBeAg chemiluminescence immunoassay (CLIA) kits were purchased from Autobio Diagnostics Co. (Zhengzhou, China). The human LDLR-expressing plasmid pCMV6-XL4/LDLR was obtained from OriGene. The infectious HBV DNA (pCMV-HBV) was described previously [59]. Lipofectamine 3000 and RNAiMax reagents were from Invitrogen.

### HBV production and concentration

HepAD38 cells were grown in DME/F12 medium containing 4% FBS and 1% dimethyl sulfoxide (DMSO) in HyperFlasks (Corning). Cell culture supernatants were collected every 6 days and were used for HBV concentration by precipitation with 10% polyethylene glycol (PEG) 8,000 (Hampton Research). The genome copy numbers of HBV DNA were determined by a real-time PCR method.

### HBV infection assay

HepG2$^{NTCP}$ Cells were seeded in 24-well cell culture plates at a density of $1.5×10^5$ per well overnight and were infected with HBV at a multiplicity of infection (m.o.i.) of 100 copies of genome equivalent in the present of 4% PEG 8000. After 12 hours infection, unbound HBV was removed by washing with 1x PBS. The HBV-infected cells were incubated with DME/F12 medium containing 3% FBS, 1% DMSO, and 5 μg/mL hydrocortisone (HC) at 37˚C for 4 days. The levels of HBcAg in the infected cells were determined by Western blot. HBV cccDNA in the cell was extracted with the Hirt method and treated with exonucleases prior to quantification by a Real-Time PCR method as described previously [22]. The levels of HBeAg and HBV

DNA in the supernatants were quantified by chemiluminescence immunoassay and qPCR, respectively.

## Blockade of HBV infection by a LDLR-specific monoclonal antibody

The LDLR-blocking monoclonal antibody (C7) was diluted to varying concentrations (0, 0.4, 2, and 10 μg/mL) with HBV in 4% PEG 8000. HBV with or without C7 was used to infect HepG2$^{NTCP}$ or PHH cells in 24-well cell culture plates. A normal mouse IgG was used as a negative control. After 12h incubation at 37˚C, the HBV-infected cells were cultured in DME/F12 medium containing 3% FBS, 1%DMSO, and 5 μg/μL hydrocortisone for 4 days. The levels of HBcAg and cccDNA in cells and HBeAg and HBV DNA in supernatants were determined as described above.

## SiRNA-mediated silencing of LDLR expression

Varying concentrations (0, 2, 10 and 50 nM) of LDLR-specific siRNAs (siLDLR) or a nonspecific control siRNA (siNSC) were transfected into HepG2$^{NTCP}$ using lipofectamine RNAiMax reagent. At 48 h post-transfection, cells were infected with HBV as described in HBV infection assay. The levels of HBcAg, cccDNA, HBeAg, and HBV DNA in the cell and supernatant were determined, respectively.

## CRISPR/Cas9-induced LDLR gene knockout in HepG2$^{NTCP}$

The LDLR gene-specific sgRNA was designed based on the methods described at the website http://crispr.mit.edu (Feng Zhang). Two synthetic oligonucleotide primers LDLR/F (5'-CACC G TCCTACAAGTGGGTCTGCGA-3') and LDLR/R (5'-AAACTCGCAGACCCACTTGTAG GA C-3') were annealed and cloned into the BsmBI-digested lentiCRISPRv2-blasticidin vector as described previously [33]. Resulting plasmid DNA pLentiCRISPRv2-Blast/LDLR-sgRNA was confirmed by DNA sequence analysis. To produce recombinant LDLR-sgRNA lentivirus, pLentiCRISPRv2-Blast/LDLR-sgRNA, pVSVg, and psPAX2 were co-transfected into HEK293T cells using Lipofectamine 3000 reagent according to the manufacturer's instructions (Invitrogen). At 3 days p.t., the supernatant was harvested and filtered through a 0.45 μm low-protein-binding membrane unit (Millipore). HepG2$^{NTCP}$ cells were transduced with the recombinant LDLR-sgRNA lentivirus and cultured in the presence of 5 μg/mL blasticidin for 2–3 weeks. Knockout of LDLR gene in stable cell clones were determined by DNA sequence and Western blot analyses, respectively.

## Ectopic expression of LDLR from a DNA vector

HepG2$^{NTCP}$ or LDLR$^{-/-}$ HepG2$^{NTCP}$ cells were seeded at $1.5 \times 10^5$ cells/well in 24-well cell culture plates. The LDLR-expressing plasmid pCMV6-XL4/LDLR DNA was transfected into cells using Lipofectamine 3000 reagent. At 48 h p.t., cells were infected with HBV as described above. The levels of HBcAg and HBV cccDNA were subsequently determined by Western blot and qPCR methods, respectively.

## Real-Time quantitative PCR (qPCR)

HBV genomic DNA in the supernatants was extracted with QIAGEN kit and HBV cccDNA in the cell was extracted by the Hirt method as described previously [17]. HBV DNA was quantified by using two HBV-specific primers: 5′-GAGTGTGGATTCG CACTCC-3′ (forward) and 5′-GAGG CGAGGGAGTTCTTCT-3′ (reverse). HBV cccDNA was quantified using primers 5′- TCATCTGCCGGACCGTGTGC-3′ (forward) and 5′- TCCCGATAC AGAGCTGA

GGCGG-3′ (reverse) and probe HBV/cccDNA-P: 5'-FAM-TTCAAGCCTCCAAGCTGTG CCTTGG GTGG C-TAMRA -3'. The qPCR was carried out under the conditions of 95˚C for 10 min (1 cycle) and 95˚C for 15s and 60˚C for 60s (40 cycles) using TaqMan SYBR Green Master Mix (Applied Biosystems) or iTaq Universal Probes Supermix (Bio-Rad) and using two HBV-specific primers as previously described [22].

## Quantification of HBeAg by chemiluminescence immunoassay

The levels of HBeAg in the cell culture supernatants were determined with a chemiluminescence immunoassay according to manufacturer's instructions, as described previously [22].

## Western blot analysis

Cells were lysed in a RIPA buffer. Protein concentrations of cell lysates were determined using protein assay dye reagent. Cell lysates of 25 μg proteins were loaded to 10–18% SDS-PAGE. Upon electrophoresis, proteins in the gel were transferred onto a polyvinylidene difluoride (PVDF) membrane using a semidry blotter (Bio-Rad). Membranes were blocked with 5% non-fat dry milk for 1 h, followed by incubation with primary antibodies specific to HBcAg, LDLR, or β-actin at 4˚C overnight. Proteins were stained with horseradish peroxidase-conjugated secondary antibodies for 1 h and subsequent ECL substrate. Protein bands were visualized and documented with ChemiDoc MP Imaging System (Bio-Rad).

## Immunofluorescence assay (IFA)

HBcAg expression in the HBV-infected cells was determined by a previously described IFA with modification [60]. Briefly, HepG2$^{NTCP}$ cells were seeded at $2 \times 10^5$ cells/well in 24-well cell culture plates with coverslips coated with collagen. Cells were infected with HBV as described above. Cells were fixed with 4% paraformaldehyde at room temperature for 15 minutes, followed by cell permeabilization with 0.1% Triton X-100 for 5 minutes and blocking with 3% BSA at 4˚C overnight. HBcAg was stained with a HBc-specific monoclonal antibody C1-5 (100-fold dilution) at 4˚C overnight and the secondary goat anti-mouse IgG conjugated with Alera Fluro 594 (2000-fold dilution) at room temperature for 2 h. Coverslips were mounted to glass slides using Prolong Diamond Antifade Mountant with DAPI (Invitrogen). Fluorescence images were taken with Keyence BZ-X800E fluorescent microscope.

## HBV cell attachment assay

HepG2, HepG2$^{NTCP}$ cells, or LDLR$^{-/-}$ HepG2$^{NTCP}$ cells were incubated with HBV at an m.o.i. of around 50 copies of genome equivalent in DME/F12 medium at 37˚C for 6 h. Unbound HBV was removed by washing with 1xPBS three times. Cell-bound HBV DNA was extracted with a QIAGEN DNA isolation kit, followed by qPCR quantification.

## Statistical analysis

Statistical analysis was performed using GraphPad Prism 5 by one-way analysis of variance followed by the Student-Newman-Keuls t-test. Results are shown as means ± standard deviations and all the data were obtained from three independent experiments. P values of ≤0.05 were considered statistically significant.

## Supporting information

**S1 Text. Supporting information.**
(DOCX)

**S1 Fig. Validation of HBV infection and replication.** HepG2$^{NTCP}$ cells seeded in 24-well plates were infected with HBV in the presence of 4% PEG. After 12-16h infection at 37˚C, the HBV-infected cells were washed with PBS and incubated with DME/F12 medium containing 3% FBS, 1% DMSO, and 5 μg/mL hydrocortisone (HC), as previously described [17]. At different time points (1 to 9 days) post-infection, cells were lysed with a RIPR buffer. **A**. Detection of HBcAg by Western blot analysis. The lysates of uninfected and HBV-infected HepG2$^{NTCP}$ cells were used for protein separation by electrophoresis in a 10% SDS-PAGE gel. HBcAg and β-actin bands were visualized by Western blotting using an HBc- and β-actin-specific antibodies. **B**. IFA immunostaining of HBcAg in the HBV-infected cells. IFA is described in material and methods. Uninfected HepG2$^{NTCP}$ cells were used as a negative control. Uninfected and HBV-infected HepG2$^{NTCP}$ cells at 16h (right after HBV incubation as a measurement of input HBV) and day 5 post-infection were subjected to immunostaining of HBcAg by IFA. The nuclei of cells (Blue) and HBcAg (Red) were stained with DAPI and anti-HBc (C1-5)/Alexa Fluor 594 goat anti-mouse IgG, respectively. Fluorescence images were taken using the Nikon S plan fluor 40x objective and either an exposure of 1/3s or 1/2s for DAPI and HBcAg, respectively. Scale bars at the bottom right corner indicate 100 μm.
(TIF)

**S2 Fig. Immunostaining of HBcAg in the HBV-infected HepG2$^{NTCP}$ cells with or without LDLR silencing.** HepG2$^{NTCP}$ cells were transfected with 50 nM of LDLR-specific siRNAs (siLDLR) or a non-specific control siRNA (siNSC) using RNAiMax (Invitrogen). At 48 hours upon siRNA transfection, cells were infected with HBV in the presence of 4% PEG at 37˚C for 16 hrs. Uninfected HepG2$^{NTCP}$ (negative control) and HBV-infected cells right after 16-h incubation (input virus control) or at day 5 post-infection were fixed with 4% paraformaldehyde, permeabilized with 0.1% Triton X-100, and stained for HBcAg by IFA. Cell nuclei were stained with DAPI (Blue) and HBcAg was stained with an anti-HBc (C1-5) and the secondary Alexa Fluor 594 goat anti-mouse IgG (red). Fluorescence images were taken using the Nikon S plan fluor 40x objective and either an exposure of 1/3s or 1/2s for DAPI and HBcAg, respectively. Scale bars at the bottom right corner indicate 100 μm.
(TIF)

## Acknowledgments

We thank Prof. Jin Ye (UT Southwestern Medical Center) for providing the LDLR monoclonal antibody HL-1 and Jason M. Needham (Sunnie Thompson's lab) for helping to take and analyze IFA images.

## Author Contributions

**Conceptualization:** Guangxiang Luo.

**Data curation:** Yingying Li, Guangxiang Luo.

**Formal analysis:** Yingying Li, Guangxiang Luo.

**Funding acquisition:** Guangxiang Luo.

**Investigation:** Yingying Li, Guangxiang Luo.

**Methodology:** Yingying Li.

**Project administration:** Guangxiang Luo.

**Supervision:** Guangxiang Luo.

**Validation:** Yingying Li.

**Writing – original draft:** Yingying Li, Guangxiang Luo.

**Writing – review & editing:** Guangxiang Luo.

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
