## [Decision Letter · Decision Letter 0]

8 Feb 2021

Dear Dr. Luo,

Thank you very much for submitting your manuscript "Human low-density lipoprotein receptor plays an important role in hepatitis B virus infection" for consideration at PLOS Pathogens. As with all papers reviewed by the journal, your manuscript was reviewed by members of the editorial board and by several independent reviewers. In light of the reviews (below this email), we would like to invite the resubmission of a significantly-revised version that takes into account the reviewers' comments. In particular, the reviewers recommended additional assays be performed to further verify HBV infection.

We cannot make any decision about publication until we have seen the revised manuscript and your response to the reviewers' comments. Your revised manuscript is also likely to be sent to reviewers for further evaluation.

Sincerely,

Jianming Hu

Associate Editor

PLOS Pathogens

Jing-hsiung James Ou

Section Editor

PLOS Pathogens

Kasturi Haldar

Editor-in-Chief

PLOS Pathogens

orcid.org/0000-0001-5065-158X

Michael Malim

Editor-in-Chief

PLOS Pathogens

orcid.org/0000-0002-7699-2064

Reviewer's Responses to Questions

**Part I - Summary**

Reviewer #1: A recent study from George Luo group reported that the cellular low density lipoprotein apolipoprotein E (ApoE) plays a significant role in HBV entry and egress through incorporating into virion particles (Qiao and Luo, Plos Pathogens 2019). ApoE interacts significantly with the low-density lipoprotein receptor (LDLR), which is essential for the catabolism of triglyceride-rich lipoproteins. In this manuscript, Luo group further investigated the function of hepatic LDLR in HBV infection and revealed the following major findings: 1) a LDLR monoclonal antibody C7 dose dependently inhibited HBV infection in both HepG2-NTCP cells and primary human hepatocytes (PHH) when being added to HBV inoculation but not after infection, indicating that blocking LDLR prevents HBV infection at early entry step(s); 2) knockdown or knockout of LDLR in HepG2-NTCP cells markedly decreased HBV infection; 3) trans-complementation of LDLR expression in LDLR-KO cells restored HBV infection; 4) C7 antibody or LDLR-depletion reduced the attachment of HBV DNA particles to HepG2-NTCP cells, and a recombinant LDLR protein blocked the binding of ApoE to heparin, indicating that LDLR acts as a HBV cell attachment receptor for HBV-associated ApoE. Overall, this study revealed a novel finding that LDLR serves as a co-factor for HBV entry in hepatocyte cells, which sheds new light on the entry mechanism of HBV entry and provides novel potential antiviral targets for development of HBV therapeutic means. The presented data are generally convincing and supportive for the major conclusions. The manuscript is also well-written in a concise manner. The authors may consider the following comments to further strengthen the manuscript.

Reviewer #2: Li and Luo report in this manuscript a very straightforward story that LDLR on hepatocytes plays an important role in HBV infection, most likely by serving as the attachment receptor for HBV virions. Overall, the studies are well designed and executed. The conclusions are supported by the data presented. However, further mechanistic insights, particularly, identification of virion component(s) directly interacts with LDLR, should improve the manuscript.

Reviewer #3: In this manuscript, Li and Luo present data in support of the human LDL receptor playing a role at the step of HBV entry in human hepatocytes. The text is well written, but the data are not convincing, in particular for the authenticity of an infection, which is documented at a too early time post inoculation. No details are given on the specificity of the cccDNA assay, and the proof that HBcAg, HBeAg and HBV DNA are markers of infection, and not derived from the inoculum, is not established. The study should include additional experiments: i) an immunofluorescence detection of HBcAg post-inoculation, and ii)a non-susceptible NTCP-negative cell line (HepG2) should be used as a control.

**Part II – Major Issues: Key Experiments Required for Acceptance**

Reviewer #1: 1. Fig. 1 and others, HBc western blot was mainly used in this study for measuring HBV infection. While the HBc western data are largely convincing to support a productive HBV infection, however, the western blot of whole cell lysate could not distinguish the intracellular HBc (internalized and de novo synthesized) from cell bound HBc in the form of inoculated viral particles, if any. Thus, although not absolutely required, an HBc immunostaining of infected cells would provide an additional way to demonstrate HBV infection, which is widely used in HBV field. In addition, Southern blotting of HBV cccDNA and core DNA is a good standard for HBV DNA detection. If possible, the authors may consider this option to make the results more compelling.

2. Fig. 7, as stated in the manuscript, the purpose of this HBV transfection experiment was to assess the potential effect of LDLR on HBV replication beyond entry events. Hence, it is better to analyze HBV core DNA rather than cccDNA. Moreover, if qPCR is used to detect HBV core DNA or cccDNA, if any, in transfection system, the input plasmid DNA should be eliminated prior to qPCR. In cytoplasmic core DNA extraction, DNase is used to remove non-encapsidated DNA in cell lysate. According to the M&M of this manuscript, only Exo I/III combo was used to treat Hirt DNA before qPCR, which was not able to remove the covalently closed plasmid DNA. Dpn I should be used to fragment plasmid DNA prior to Exo I/III digestion. However, to the best knowledge of this reviewer, a qPCR method for detecting HBV cccDNA in transfected cells has not been reported with validation thus far, and the best evidence supporting cccDNA production in HBV plasmid transfection is the Southern blot data from Jianming Hu’s group. If Southern blotting assay is not available to the authors, I would suggest the authors show the core DNA qPCR instead of cccDNA qPCR. Lastly, pCMV-HBV plasmid was used in this experiment, which could not rule out a potential effect of LDLR on HBV core promoter-based transcription. Thus, pHBV1.3 plasmid, which transcribes pgRNA under the authentic HBV core promoter, would serve the authors’ purpose better.

3. Fig. 8, the virus-cell attachment assay is normally done under 4°C to block virus internalization, please explain why 37°C was used in this study. In addition, the HepG2 cells (without NTCP) can serve as a clean target cell to specifically assess the binding of HBV to cell upon blocking or depleting LDLR without worrying about NTCP-mediated HBV endocytosis.

Reviewer #2: 1. There are two alleles of LDLR gene for each cell and the sgRNA targeted sequence for only one allele from each cell clone is represented in Figure 5A. How about another one? It is unlikely that the two alleles are repaired exactly the same way.

2. Figure 7. in the transient transfection assay, almost all the HBcAg should be translated from pgRNA transcribed from input plasmid DNA and do not reflect the activity of viral DNA replication. In addition, due to the presence of large amount of plasmid DNA, PCR quantification of cccDNA under this experimental condition is difficult and also not necessary. The best way to measure HBV replication in this experiment should be detecting viral DNA replication intermediates by Southern blot hybridization.

3. For the experiments presented in Figure 8, it will be interesting to see if over-expression of LDLR in HepG2-NTCP cells dose-dependently enhances the attachment of HBV.

4. In order to distinguish attachment/binding and internalization of viral particles during infection, the binding assay is usually done at 4 °C to prevent endocytosis or membrane fusion. The current assay condition (37 °C) actually cannot distinguish whether LDLR enhances attachment/binding or internalization. In my point view, this later possibility cannot be ruled out.

5. Does purified recombinant ApoE block HBV attachment/binding or infection?

Reviewer #3: i) an immunofluorescence detection of HBcAg post-inoculation - at 9-12 dpi.

ii) a non-susceptible NTCP-negative cell line (HepG2) should be used as a control.

**Part III – Minor Issues: Editorial and Data Presentation Modifications**

Reviewer #1: 1. Page 15, the sentence “The genome copy numbers of HBV DNA were determined by a real-time PCR method.” needs information of qPCR primers/probe or a reference.

2. Fig. 1A, would pre-incubation of cells with LDLR antibody achieve a better inhibition of HBV infection?

3. Fig.3B, please specify when C7 was added after infection. The loss of inhibition of HBcAg production by C7 after HBV infection actually indicates that there is literally no virus spread in HepG2-NTCP cells, the authors may consider to mention this point.

4. Fig. 4, it seems that 2nM siLDLR reduced HBV DNA and HBeAg more obviously than HBcAg. A densitometric analysis of the western bands may help to show the protein levels quantitatively.

5. Fig. 9 needs a non-ApoE negative control.

Reviewer #2: 6. “HBV cccDNA in the cell was extracted by the Hirst method as described previously”, it should be Hirt Method.

Reviewer #3: (No Response)

PLOS authors have the option to publish the peer review history of their article (what does this mean?). If published, this will include your full peer review and any attached files.

Reviewer #1: No

Reviewer #2: No

Reviewer #3: No
---

## [Decision Letter · Decision Letter 1]

11 May 2021

Dear Dr. Luo,

Thank you very much for submitting your manuscript "Human low-density lipoprotein receptor plays an important role in hepatitis B virus infection" for consideration at PLOS Pathogens. As with all papers reviewed by the journal, your manuscript was reviewed by members of the editorial board and by several independent reviewers. The reviewers appreciated the attention to an important topic. Two reviewers have no more issues with the revised manuscript. However, reviewer 3 has some remaining concerns on the validation of infection. Based on the reviews, we are likely to accept this manuscript for publication, provided that you modify the manuscript to address reviewer 3's concerns. 

Sincerely,

Jianming Hu

Associate Editor

PLOS Pathogens

Jing-hsiung James Ou

Section Editor

PLOS Pathogens

Kasturi Haldar

Editor-in-Chief

PLOS Pathogens

orcid.org/0000-0001-5065-158X

Michael Malim

Editor-in-Chief

PLOS Pathogens

orcid.org/0000-0002-7699-2064

Reviewer Comments (if any, and for reference):

Reviewer's Responses to Questions

**Part I - Summary**

Reviewer #1: The authors have addressed my previous comments adequately, the manuscript has been further strengthened by providing more evidence to support the involvement of LDLR in HBV infection.

Reviewer #2: The authors addressed my concerns on the previous version with satisfaction.

Reviewer #3: This is a very well written manuscript that presents experiments conducted to demonstrate the role of the low-density lipoprotein receptor (LDLr) in HBV infection (viral entry). This is a follow-up of a previous study from George Luo's group (doi.org/10.1371/journal.ppat.1007874) that described the role apolipoprotein E in assembly and infectivity of HBV virions. If confirmed, the data presented in this manuscript represent an important new finding in the mechanism of HBV entry. It is thus important that experiments be more convincing by using several methods for proof of infection. To this reviewer's opinion, the detection of HBcAg by immunoblotting is not sufficient as a proof of infection. qPCR assay for specific detection of cccDNA is also controversial for its tendency to react with rcDNA. It might also be argued that HBcAg signals be derived from the inoculum, especially when considering that capsids and empty virions (virions with empty capsids) are present in AD38 supernatants at an order of magnitude 2 logs above that of DNA containing virions. This ratio also depends upon culture conditions.

**Part II – Major Issues: Key Experiments Required for Acceptance**

Reviewer #1: (No Response)

Reviewer #2: No.

Reviewer #3: To improve the quality of this study, the authors should more precisely describe the HBV particles used for infection. The particles were precipitated with 10% PEG, which is known to also precipitate non-enveloped capsids and nucleocapsids. The virus preparations should be characterized for the presence of LDL, Apo B, Apo A etc. and it should be shown that what is at the origin of HBcAg detection post inoculation does not result from a binding of capsids or nucleocapsids via lipoproteins. Different sources of inoculum would also be required to demonstrate a general phenomenon.

It is equally important that infections be confirmed using immunofluorescence assay for detection of intracellular HBcAg and/or HBsAg in order to measure a percentage of infected cells.

To unequivocally proof infection, Northern blotting should be performed as it will detect an HBV product (mRNAs) that is not present in the inoculum.

This reviewer would fully agree with the conclusion of this manuscript if results are confirmed by the suggested experiments.

**Part III – Minor Issues: Editorial and Data Presentation Modifications**

Reviewer #1: (No Response)

Reviewer #2: No.

Reviewer #3: (No Response)

PLOS authors have the option to publish the peer review history of their article (what does this mean?). If published, this will include your full peer review and any attached files.

Reviewer #1: No

Reviewer #2: No

Reviewer #3: No

Figure Files:

Data Requirements:

Reproducibility:

References:

---

## [Editor Report · Decision Letter 2]

17 Jun 2021

Dear Dr. Luo,

We are pleased to inform you that your manuscript 'Human low-density lipoprotein receptor plays an important role in hepatitis B virus infection' has been provisionally accepted for publication in PLOS Pathogens.

Should you, your institution's press office or the journal office choose to press release your paper, you will automatically be opted out of early publication. We ask that you notify us now if you or your institution is planning to press release the article. All press must be coordinated with PLOS.

Best regards,

Jianming Hu

Associate Editor

PLOS Pathogens

Jing-hsiung James Ou

Section Editor

PLOS Pathogens

Kasturi Haldar

Editor-in-Chief

PLOS Pathogens

orcid.org/0000-0001-5065-158X

Michael Malim

Editor-in-Chief

PLOS Pathogens

orcid.org/0000-0002-7699-2064
---

## [Editor Report · Acceptance letter]

19 Jul 2021

Dear Dr. Luo,

We are delighted to inform you that your manuscript, "Human low-density lipoprotein receptor plays an important role in hepatitis B virus infection," has been formally accepted for publication in PLOS Pathogens.

Best regards,

Kasturi Haldar

Editor-in-Chief

PLOS Pathogens

orcid.org/0000-0001-5065-158X

Michael Malim

Editor-in-Chief

PLOS Pathogens

orcid.org/0000-0002-7699-2064